# Fabrication and Investigation of Graphite-Flake-Composite-Based Non-Invasive Flex Multi-Functional Force, Acceleration, and Thermal Sensor

**DOI:** 10.3390/mi14071358

**Published:** 2023-06-30

**Authors:** Noshin Fatima, Khasan S. Karimov, Farah Adilah Jamaludin, Zubair Ahmad

**Affiliations:** 1Faculty of Engineering, Technology and Built Environment, UCSI University, Kuala Lumpur 56000, Malaysia; farahadilah@ucsiuniversity.edu.my; 2Ghulam Ishaq Khan Institute of Engineering Sciences and Technology, Topi 23640, Pakistan; khasan@giki.edu.pk; 3Center for Innovative Development of Science and Technologies of Academy of Sciences, Dushanbe 734025, Tajikistan; 4Qatar University Young Scientists Center (QUYSC), Qatar University, Doha 2713, Qatar; zubairtarar@qu.edu.qa

**Keywords:** biodegradable, graphite nanoflakes composite, carbon compound electronics, environmental education, international poverty line, pollution-free, micro-technology

## Abstract

This work examines the physics of a non-invasive multi-functional elastic thin-film graphite flake–isoprene sulfone composite sensor. The strain design and electrical characterization of the stretching force, acceleration, and temperature were performed. The rub-in technique was used to fabricate graphite flakes and isoprene sulfone into sensors, which were then analyzed for their morphology using methods such as SEM, AFM, X-ray diffraction, and Fourier transform infrared spectroscopy to examine the device’s surface and structure. Sensor impedance was measured from DC to 200 kHz at up to 20 gf, 20 m/s^2^, and 26–60 °C. Sensor resistance and impedance to stretching force and acceleration at DC and 200 Hz rose 2.4- and 2.6-fold and 2.01- and 2.06-fold, respectively. Temperature-measuring devices demonstrated 2.65- and 2.8-fold decreases in resistance and impedance at DC and 200 kHz, respectively. First, altering the graphite flake composite particle spacing may modify electronic parameters in the suggested multi-functional sensors under stress and acceleration. Second, the temperature impacts particle and isoprene sulfone properties. Due to their fabrication using an inexpensive deposition technique, these devices are environmentally friendly, are simple to build, and may be used in university research in international poverty-line nations. In scientific laboratories, such devices can be used to teach students how various materials respond to varying environmental circumstances. They may also monitor individuals undergoing physiotherapy and vibrating surfaces in a controlled setting to prevent public health risks.

## 1. Introduction

The manufacturing and investigation of devices for measuring force [1,2], acceleration [3,4], and temperature [5,6] are crucial for two reasons: first, for practical applications, and second, for the development of relevant technology in the near future of biotechnology [7]. Based on cost and impedance matching, a systematic method was devised for selecting force and acceleration sensors [8,9,10,11] based on their electrical and mechanical qualities. In [12], the authors presented a graphite-based flexible piezoresistive device on a paper substrate. The manufacture of the gadget was a straightforward and quick process. In [13], an accelerometer was fabricated by using paper coated with piezoelectric ZnO nanowires as a base. Other researchers discussed ways to modify the force control of accelerometers and explored how it could be used near the end-effectors of the robot’s manipulators. They introduced an innovative method that relies on the observation-based technique by utilizing the fusion of different sensors, and the sensor was introduced into a robot-grinding tool [14]. In addition, some wildlife motion sensors were proposed in [15] to observe the activities of the desert tortoise. Another group of researchers introduced a sensor selection scheme for controlling the impedance. These sensors should be selected by matching their properties to the requirements of the application. A flexible piezoresistive MEMS force sensor was fabricated on paper as a substrate material in [16]. The results showed moderate performance when measuring the force; however, it was also used under a balanced weight with a 25 mg resolution and a 15 g measurement range. On the principle of electromagnetic induction, another angular acceleration sensor was designed and fabricated with an innovative structure in Ref. [17]. The cited work mentioned that the sensor excitation windings were constructed under a constant magnetic field; however, for cutting the cup-shaped magnetic fields, a rotor was used. The electromotive force (EMF) was generated as an output of the sensor’s windings. However, the EMF has a direct relation with angular acceleration (ω) due to the rotational angular acceleration of the rotor produced as a result of electromagnetic coupling. It was shown in [18] that a silicon-based accelerometer could be made flexible by connecting its piezoresistors in a Wheatstone bridge configuration, resulting in a differential output that described the semiconductor mass supported by the frames. Different researchers [19,20,21,22] have detailed the design, manufacture, and use of flexible strain sensors for the in situ measurement of parachute canopy deformation and other applications, including the analysis of textile structures based on a conductive polymer composite in [23]. The bending impact on flexible thin-film transistors was investigated. Other methods for studying the dynamics of a falling item in the lab while subjected to gravity’s acceleration (g) are discussed in [24]. An infrared (IR) transceiver was used to determine g, and the reflected IR intensity from a falling rod was relayed to a digital oscilloscope so that the time of descent could be recorded. In addition, g was determined to be 9.8092 ± 0.0384 m per second squared by fitting the data to the standard quadratic equation of motion under uniform acceleration. To counter this, A. Din et al. [25] constructed multi-functional nanocomposite-based sensors with electrical characterization and improved humidity and temperature sensitivities.

Last year, we fabricated and investigated several electro-mechanical devices. Graphene was used as a composite with bismuth telluride by N. Fatima et al., and its thermoelectric properties were better than a CNT composite with bismuth telluride [26]. Recently, temperature sensors were utilized to measure the physiological states of sheep neck, tail, and under-the-tail thermal strain [27]. In [28,29,30], the authors fabricated and investigated strain gauges for measuring displacement. Carbon nanotube (CNT) composites for flexible impedance and capacitance tensile load sensors were reviewed [31], and innovative CNT-based pressure and displacement sensors were described [32]. For telemetry system applications, the researchers looked at displacement-sensitive organic field-effect transistors [33] and analyzed the transverse tensile resistive effect in TEA(TCNQ)_2_ crystals [34]. In [35], 2D materials and colloidal quantum dot graphene-based infrared photodetectors were used as CMOS for light harvesting. It has a gate-tunable ambipolar feature gate bias <3.3 V and can detect spectra from visible to near-infrared light with a gain of up to 10^5^ and a response time of 3 ms. The phototransistor has 10^4^ AW^−1^ responsivity and 10^12^ Jones-specific detectivity (at 1550 nm at 1 V).

As the literature presents that carbon has different applications, in this study, we were more focused on graphite. In simple terms, graphene is a monolayer of graphite, a mineral ubiquitous in nature, and graphite flakes comprise numerous layers of graphene. Hence, in the present study, we opted to utilize graphite flakes owing to the presence of numerous graphene layers, which provides an opportunity for enhancing the responsivity of devices across a diverse range of force, acceleration, and thermal conditions. Furthermore, in this study, we report the fabrication and analysis of elastic sensors that measure stretching force, acceleration, and temperature and are based on a graphite flake–isoprene sulfone composite, which is part of our ongoing effort to fabricate and investigate electronic devices based on organic materials. 

## 2. Materials and Methods

For the fabrication of multi-functional elastic thin-film sensors for stretching force, acceleration, and temperature, commercially available graphite flakes were purchased from Sigma Aldrich, CAS Number 1034343-98-0, with an electronic conductivity of 10^3^ S/m and a lateral size of 0.5–5 μm. A 100 μm thin layer of ²-isoprene sulfone (C_5_H_8_O_2_S) isoprene sulfone film was used as the substrate, and graphite flake powder was utilized as received without any modification. For the fabrication of devices, the isoprene sulfone film was adhered to the solid platform under tension and at room temperature, graphite flakes were deposited using pressing-sliding (or rubbing-in) technology, and the film was constructed on the substrate, as depicted in Figure 1a.

During the rubbing-in process for device fabrication, the constant pressure over the graphite nanoflake powder was 25–35 gf/cm^2^. As a result, solid graphite flake–isoprene sulfone composite films with a thickness of 10–20 μm were formed. To observe their surface morphology, atomic force microscopy was performed by using the FLEX-AFM model, manufactured by NANOSURF. The graphite flake–isoprene sulfone surface roughness is 0.295 µm in Figure 1b. However, Figure 1c presents the scanning electron microscopy (SEM) result of the sample, which was obtained using a Phillips XL30 microscope. Figure 1b,c show the morphological results of graphite flakes on an isoprene sulfone sample at 5 microns, which clearly shows that the graphite flakes are settled within the porous surface of isoprene sulfone. However, heat production over isoprene sulfone causes the graphite flakes to form a composite with isoprene sulfone. 

A schematic diagram of the multi-functional thin graphite flake–isoprene sulfone composite film sensor for stretching force, acceleration, and temperature is shown in Figure 2. The sensor consists of an elastic (metallic spring) cell, an elastic thin graphite flake–isoprene sulfone composite film, conductive terminals, and weights, which were used to calibrate the sensor. The total length and width of the elastic thin isoprene sulfone film were equal to 6 cm and 1 cm, respectively. In the initial stage, under the effect of the spring, the elastic thin graphite flake–isoprene sulfone composite film was in tension with relatively low resistance.

## 3. Results and Discussion

The results of the fabricated sensors were analyzed for their morphological and electrical properties using XRD, FTIR, force, acceleration, and temperature measurements at different frequencies. The XRD analyzer identified the crests of graphite flakes at 26.47° (highest peak), 43.4°, 44.55°, 54.51°, and 77.6°, with associated crystal plane diffractions of (002), (100), (101), (004), and (110), corresponding to JCPDS card no. 00-041-1487, as presented in Figure 3. The unit cell of graphite flakes has the following volume, lattice constant, d-spacing, crystallite size, and microstrain values: 35.13 Å^3^, 0.68 Å, 3.43 Å, 9.435 nm, and 18.202 × 10^−3^ nm, respectively.

The absence of prominent diffraction peaks in amorphous materials may be due to the destruction of the crystal structure, which is logical given that these types of molecular crystals are typically generated by van der Waals bonding. However, the isoprene sulfone X-ray diffraction spectrum shows amorphous polymers with wide peaks. Figure 3 shows the crystalline phase of isoprene sulfone with a medium peak intensity at 2θ of around 19° and a crystal plane of (011). The isoprene sulfone diffractogram shows an amorphous phase with a peak widening at 2θ of about 30.4° with a crystalline phase of (−211), according to JCPDS card no. 00-022-1747.

Figure 4 depicts the results of graphite flakes and isoprene sulfone. In the functional group region of the FTIR spectrum of graphite flakes, a broad, strong peak at 3420 cm^−1^ represents O-H stretching of an alcohol group via an intermolecular bond. However, the C-H stretching of an alkane group is observed due to a medium peak at 2921 cm^−1^ and a faint peak at 2855 cm^−1^. The C=C elongation of the disubstituted (cis) alkene is inferred from the band at 1631 cm^−1^. An acute peak at 1491 cm^−1^ shows C-H bending, a broad medium peak at 1096 cm^−1^ demonstrates anhydride CO-O-CO stretching, and a faint peak at 742 cm^−1^ reveals disubstituted C=C bending.

In the case of isoprene sulfone, the slight peaks at 3778 cm^−1^ and 3699 cm^−1^ imply the stretching of the O-H groups of an alcohol, whereas the medium peaks at 2925 cm^−1^ and 2885 cm^−1^ reflect the stretching of the C-H groups of an alkane. The faint, broad peak of symmetric and asymmetric stretching vibrations at 1741 cm^−1^ is attributable to CH_2_ and CH_3_ bending. C-H bending vibrations in the methyl group are detected at 1487 and 1459 cm^−1^. At 1349 cm^−1^, a moderately broad peak represents the S=O fingerprint of sulfone. A medium peak at 1191 cm^−1^ indicates the C-O stretching of a secondary alcohol, while a robust peak at 1143.27 cm^−1^ demonstrates the C-O stretching of an aliphatic ether. A strong peak at 856.87 cm^−1^ and a medium peak at 680 cm^−1^ represent C-H 1,2,4-trisubstituted bending and a benzene derivative, respectively.

The multi-functional thin-film sensor used for force measurements (Figure 2) was also utilized for acceleration measurements. Acceleration (a) was calculated using Newton’s second law of motion. Force is represented by *F*, and *m* represents the sum of the original fixed mass plus the variable mass as appropriate. If the sensor is loaded with variable mass that increases gradually, then accordingly, there will be an increase in the resistance of the sensor due to the stretching effect. A change om the total mass allows us to obtain different values of the sensor’s resistance, which should be proportional to the values of acceleration when it is measured directly. The crystalline phases and surface morphology of the sensors were investigated using a Bruker D8 Advance model X-ray diffractometer (for powder and pellets) in Bragg–Brentano (θ–2θ) scanning mode. Fourier transform infrared spectroscopy (FTIR) analysis (Perkin Elmer, Frontier) equipped with a Pike Technologies GladiATR probe in the range of 400–4000 cm^−1^ was used to identify the functional groups present in the pristine flake graphite and isoprene sulfone. For measurements of the temperature, TECPEL 322 was used, and resistance and impedance were measured with an MT 4090 LCR meter. The force was created with the use of calibrated weights.

The working phenomena of the multi-functional sensor for the stretching force are due to a mechanism in which, under the effect of the stretching force on the sensor, elastic deformation takes place along the sensor’s length and width. However, due to this stretching, the length increases, and there is a decrease in the width. This results in an increment in the resistance of the device. The calibration of the sensor with standard weights (in the range of 0–20 gf) allowed us to estimate unknown weights in the range of the investigated values of weight. Figure 5 shows the experimental characterization of the stretching force in relation to resistance and impedance. 

It is seen in Figure 5 that as the stretching force (P) increases, the resistance and impedance are increased as well. The graph slopes (S), which represent the sensitivity of the sensor, can be estimated by determining the following coefficients:(1)$$S_{R}=\frac{\mathrm{dR}}{\mathrm{RdP}}$$
(2)$$S_{Z}=\frac{\mathrm{dZ}}{\mathrm{ZdP}}$$
where impedance is denoted by Z, while resistance is denoted by R. These coefficients illustrate the level of sensitivity possessed by the force sensors. The slopes were found to be equivalent to 9.76 × 10^−3^ gf^−1^, 6.25 × 10^−3^ gf^−1^, 5.36 × 10^−3^ gf^−1^, 15.96 × 10^−3^ gf^−1^, 16.17 × 10^−3^ gf^−1^, and 16.18 × 10^−3^ gf^−1^, respectively, at DC, 100 HZ, 1 kHz, 10 kHz, 100 kHz, and 200 kHz. These values were derived using the appropriate frequency ranges. Instead of changes in the fundamental characteristics of the graphite flake particles that make up the graphite flake–isoprene sulfone composite, the relationships that are shown in Figure 5 may be explained, from a purely physical standpoint, by an increase in the distance that exists between individual graphite flake particles in the composite as a result of the stretching force. Due to the presence of the capacitive component in the impedance, the contribution of reactive conduction increases with an increase in frequency. This is probably the reason for the decrease in impedance if we consider the equivalent circuit of the samples as a parallel connection of resistance and capacitance.

The stretching force and impedance, for example, at 100 Hz, can be described by the following expression based on experimental data:(3)$$Z\left(100\right)=0.25\left(\frac{M\Omega }{\mathrm{gf}}\right)*P\left(\mathrm{gf}\right)+40\left(M\Omega \right)$$

Specifically, the effect of strength was studied in graphene oxide/polyvinyl alcohol composites, which were proposed as promising biomaterials for biomedical and tissue engineering, but whose poor mechanical and water retention properties hampered their development [36]. Nevertheless, a novel graphene-based optical micro-electro-mechanical system (MEMS) accelerometer dependent on the intensity modulation and optical properties of graphene was developed [37]. The sensing system of this MEMS accelerometer comprised a multi-layer graphene finger, a laser diode, a light source, a photodiode, and integrated optical waveguides. On the basis of simulation results, the following functional characteristics were determined: mechanical sensitivity of 1019 nm/g, optical sensitivity of 145.7%/g, resonance frequency of 15,553 Hz, bandwidth of 7 kHz, and measurement range of ±10 g. 

In addition to force, acceleration measurement is widely employed in practice. In anticipation of the advancement of accelerator technology, we designed, manufactured, and tested a sensitive accelerometer with simple symmetry. The results are depicted in Figure 6.

The experimental results demonstrate that as acceleration increases, so does impedance. The likely reason for the increase in impedance with an increase in acceleration is the same as in the previous scenario. The acceleration is directly proportional to the force; therefore, the force of stretching will increase the distance between graphite flake particles in the composite and, consequently, the impedance. Therefore, the impedance will increase due to an increase in resistive components and a decrease in capacitive components, resulting in an increase in impedance. As a result, the frequency increases, and the capacitive component of the impedance causes the impedance to decrease. The slopes (S) of the graphs can be approximated using the following coefficients:(4)$$S_{Z}=\frac{\mathrm{dZ}}{\mathrm{ZdA}}$$
where “A” is the acceleration.

It is observed in Figure 6a–c that the slopes that characterize the sensitivity are equal to 0.0167 s^2^/m, 0.00433 s^2^/m, and 0.00648 s^2^/m at frequencies of 200 kHz, 100 kHz, and 100 Hz. Similar to Expression (3), the acceleration–impedance relationship can represent the sensitivity of the sensor as follows:(5)$$Z\left(100\right)=0.28\left(\frac{M\Omega \cdot s^{2}}{m}\right)*A\left(\frac{m}{s^{2}}\right)+40\left(M\Omega \right)$$

It is seen in Figure 7 that the increase in temperature in the range of 27–60 °C causes a decrease in the impedance. The temperature sensitivity coefficients of the impedance can be estimated by the following relationship:(6)$$S_{Z}=\frac{\mathrm{dZ}}{\mathrm{ZdT}}$$

It was found that the initial (at 27 °C) and final (at 60 °C) temperature coefficients of impedance are equal to the following values at 100 Hz: −0.0583 C^−1^ and −0.025 C^−1^. However, the same sensor sensitivity at 1 kHz is equal to −0.0595 C^−1^ and −0.0208 C^−1^. It is seen that with the increase in temperature, the temperature coefficients decrease by more than two times; however, with an increase in the frequency, the changes in the concerned coefficients are significantly smaller. At higher frequencies (10 kHz, 100 kHz, and 200 kHz), the changes in the impedance values with an increase in temperature are almost negligible. Thus, the temperature dependence on the resistance and impedance of the graphite flake–isoprene sulfone composite is due to the particular properties of the composite of graphite flakes and isoprene sulfone. Furthermore, a similar expression can describe the experimental data in Figure 7 related to impedance and temperature relationships as follows:(7)$$Z\left(100\right)=0.7\left(\frac{M\Omega }{{}^{\circ}C}\right)*T\left({}^{\circ}C\right)+40\left(M\Omega \right)$$

Expressions (3), (5), and (7) represent the obtained experimental results; however, for a simulation, we can use an approximation of the device as a linear function. Previous research examined the temperature dependence of graphite and graphite composites. It was reported in [38] that graphite nanoplatelets (GNPs) were utilized to fabricate a temperature sensor. It was observed that the variation in resistance with temperature in the range of 10–60 °C was linear, making it suitable for sensor applications. TCR = 0.0371. The obtained TCR value is lower in the low-temperature region and greater in the high-temperature region. In addition, the TCR was positive in [38]. In our case, however, the result is negative. This is unquestionably attributable to differences in the composition and structure of our samples, as described in [38]. Similarly, both outcomes surpass those of the multi-walled carbon nanotubes presented in [39] as well. Even though a small amount of hysteresis was observed, it was determined that the sensors have potential applications as highly sensitive and rapid-response temperature sensors [39]. Graphite is currently used for drug testing in biochemical sensors and gas monitoring [40].

The investigation of elastic thin-film sensors based on a graphite flake–isoprene sulfone composite for measuring force, acceleration, and temperature enables the fabrication of multi-functional sensors used in numerous areas of instrumentation technology. Reference [41] describes ubiquitous, multi-functional, printed graphite sensors. Wearable resistive bending sensors that have been devised can be used to measure force, deflection, and curvature. In the review in [42], measurements of a number of parameters, including body temperature, heart rate, pulse oxygenation, respiration rate, blood pressure, blood glucose, electrocardiogram signals, electromyogram signals, and electroencephalogram signals, were presented. These variables are essential for estimating health conditions. In Reference [42], the most recent graphite-based sensors for health monitoring, novel structures, sensing mechanisms, technological advancements, sensor system components, and potential challenges are discussed. In [43], multi-functional applications of graphite-based laser-induced sensors were presented. Commercial polymer layers were laser-irradiated and applied for the photo-thermal generation of graphite for the fabrication of capacitive sensors for strain and electrochemical parameters. However, a reduced oxide graphite (rGO)-based wearable multi-functional sensor was fabricated in [44], where rGO was deposited over a porous inverse opal acetyl cellulose (IOAC) substrate. It was used for monitoring human motion and ion concentrations in sweat, with the strain-sensing layer detecting resistance changes due to human motion. It was found that the devices were able to monitor bending, rotation, and other small-scale human motions of body parts, such as the finger, wrist, head, and throat. In [45], the authors presented an investigation of recent developments in flexible pressure sensors, trends in E-skin and wearable flexible sensors, and multi-functional equipment; exploring new sensing mechanisms, seeking new functional materials, and developing novel integration technologies for flexible devices will be key directions in the sensor field in the future. 

The information presented in this paper shows that, at present, multi-functional graphite-based sensors cover a very wide range of devices for the measurement of environmental, industrial, and technological parameters [36,37,38,39,40,41,42], and this process will continue. We fabricated flexible thermoelectric devices in [46] by using rubbing-in technology based on a rubber–graphite composite, and fascinating results were obtained. In [25], we used a bulky elastic layered rubber–graphite composite to investigate the multi-sensing properties of devices fabricated using rubbing-in technology. The effects of temperature, compressive displacement, pressure, and humidity on the electric parameter, i.e., impedance, were investigated at varying frequencies in the range of 0–200 kHz. There was a decrease in impedance under the effect of uniaxial compressive displacement and pressure. However, the observed temperature coefficients were −0.836 and −0.862 (%/°C) with temperature increases from 29 °C to 54 °C and a decrease in impedance of 1.26 ± 0.01 times, respectively. Surprisingly, the device was not affected by changes in the relative humidity in the range of 58–93 %RH [25].

A comparison of the published data presented in this paper revealed that the experimental results can be used for the practical application of the sensors under investigation. Due to the use of graphite flakes and a thin isoprene sulfone film as a substrate, the sensors are sufficiently sensitive, easy to fabricate, inexpensive, and reliable for use in practice.

## 4. Conclusions

In this paper, as a continuation of our efforts in sensor design, fabrication, and investigation, the physical properties of elastic thin-film sensors for measuring tensile force, acceleration, and temperature are described. The investigation of sensors based on a graphite flake–isoprene sulfone composite revealed that the properties of the samples depend not only on the composition but also on the fabrication technology, structure, and thickness of the samples, which can be utilized in practical applications of multi-functional sensors. Due to their inexpensive deposition technology, these devices are safe for the environment, simple to produce without squandering energy, and able to play an essential role at the university level for research purposes in countries below the international poverty line. These instruments have pedagogical value in science laboratories, where students can observe how various substances react to varying environmental conditions. In addition, they have a wide range of applications in the medical field, where they can be used to monitor patients undergoing physiotherapy and to keep vibrating surfaces secure in a controlled environment to prevent adverse effects on public health.

## Figures and Tables

**Figure 1 micromachines-14-01358-f001:**
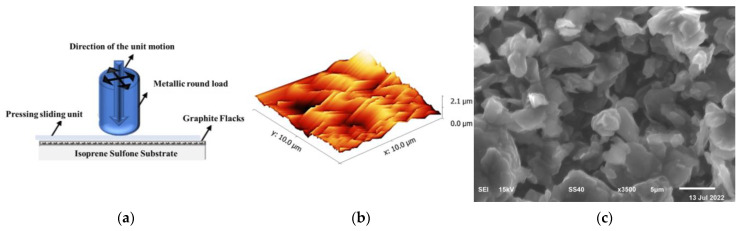
(**a**) Fabrication of the isoprene sulfone–graphite flake composite samples by pressing-sliding/rubbing-in technology, (**b**) surface morphology of graphite flake–isoprene sulfone composite film in 3D AFM image, and (**c**) SEM image.

**Figure 2 micromachines-14-01358-f002:**
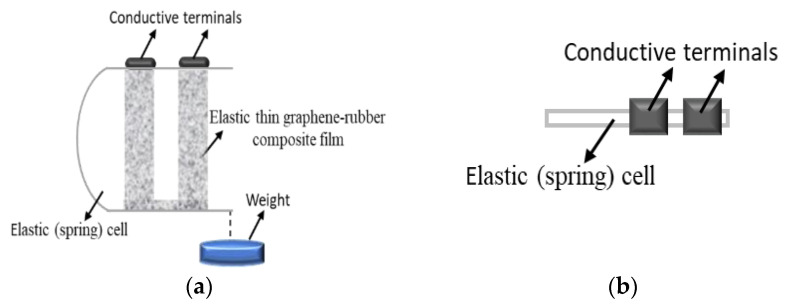
Schematic diagram of the thin flexible graphite flake–isoprene sulfone composite multi-functional sensor for measuring stretching force, acceleration, and temperature: side view (**a**) and top view (**b**).

**Figure 3 micromachines-14-01358-f003:**
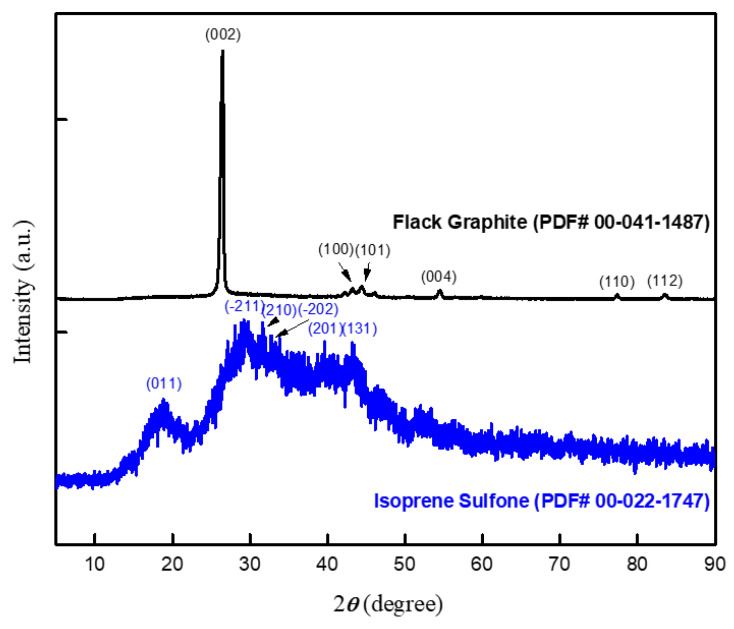
X-ray diffraction of graphite flakes (black color) and isoprene sulfone (blue color) pristine samples.

**Figure 4 micromachines-14-01358-f004:**
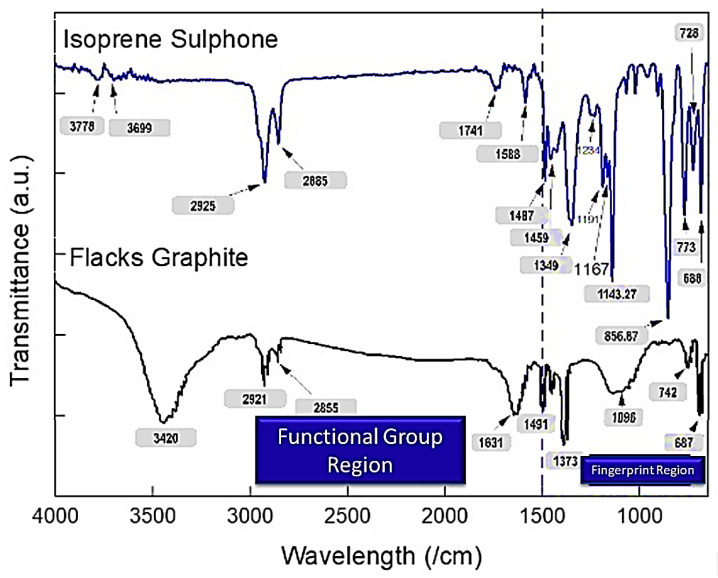
FTIR of pristine samples of graphite flakes (black color) and isoprene sulfone (blue color).

**Figure 5 micromachines-14-01358-f005:**
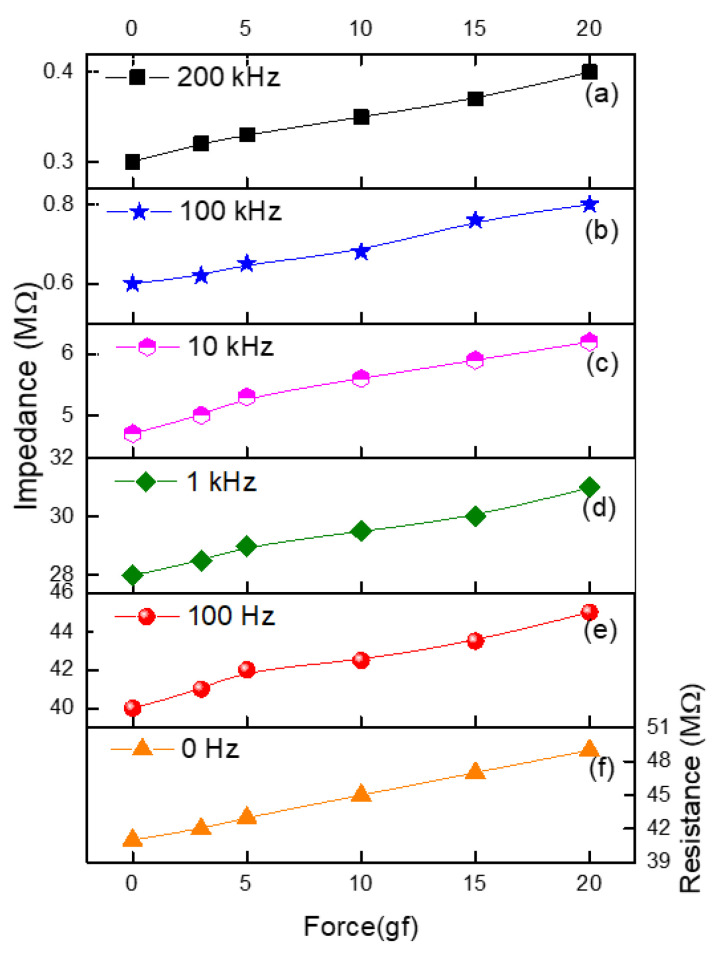
Stretching force in relation to the impedance at (**a**) 200 kHz, (**b**) 100 kHz, (**c**) 10 kHz, (**d**) 1 kHz, and (**e**) 100 H, and (**f**) resistance relationships obtained experimentally at direct and alternating currents at different frequencies.

**Figure 6 micromachines-14-01358-f006:**
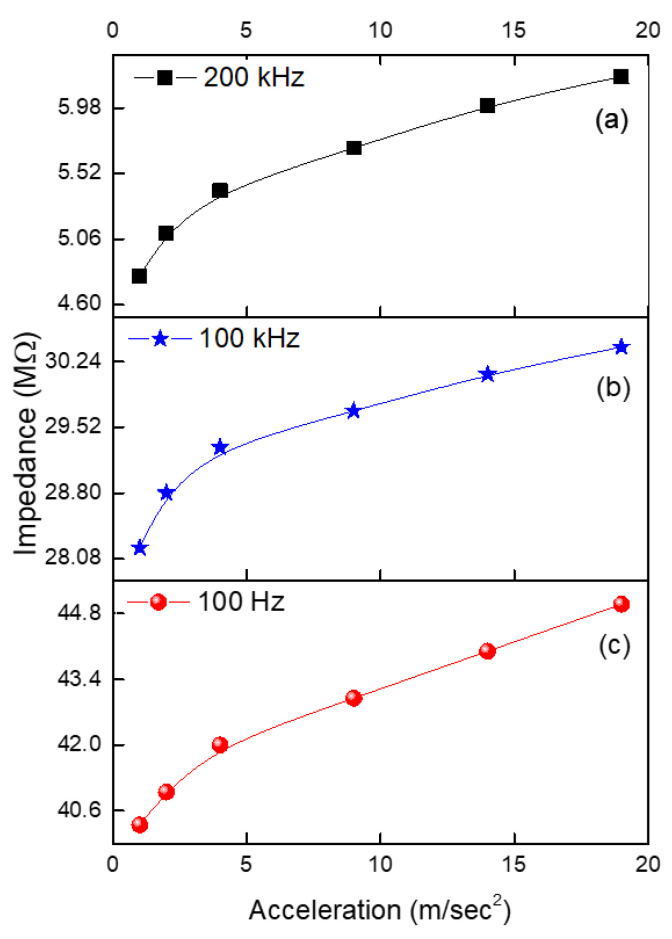
Shows the acceleration–impedance relationships at different frequencies: (**a**) 200 kHz, (**b**) 100 kHz, and (**c**) 100 Hz.

**Figure 7 micromachines-14-01358-f007:**
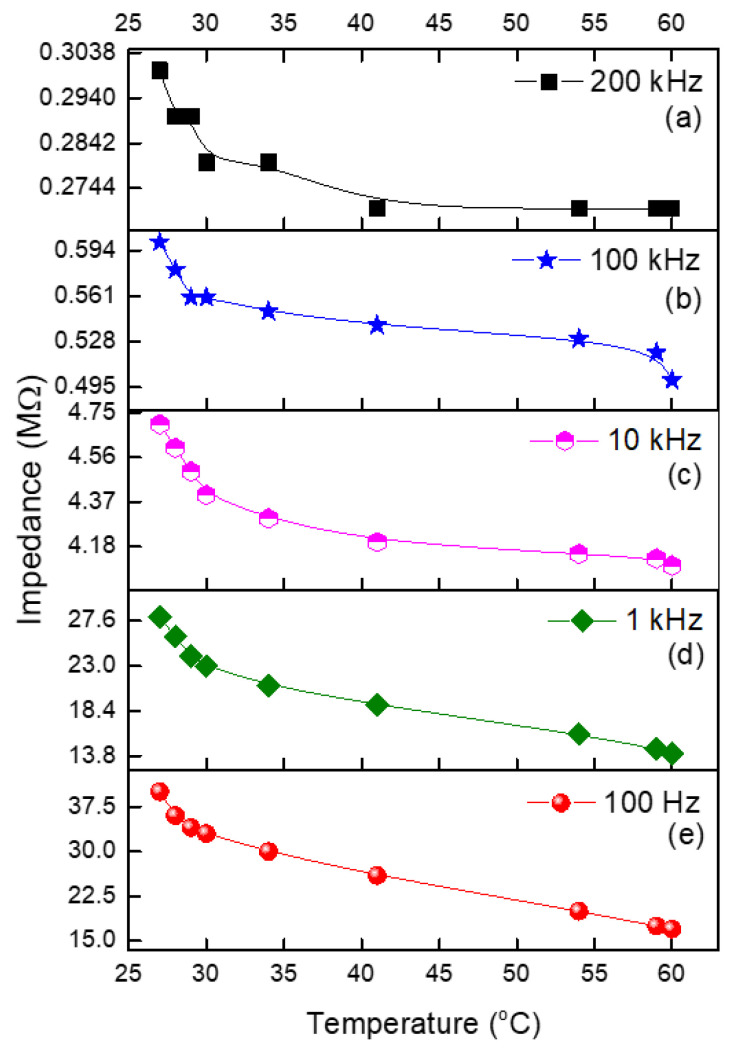
Dependence of the impedance at (**a**) 200 kHz, (**b**) 100 kHz, (**c**) 10 kHz, (**d**) 1 kHz, and (**e**) 100 Hz on temperature.

## Data Availability

The data that support the findings of this study are available within the article.

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
