# Peer review of "Fabrication and Investigation of Graphite-Flake-Composite-Based Non-Invasive Flex Multi-Functional Force, Acceleration, and Thermal Sensor"

_micromachines, 2023, doi:10.3390/mi14071358_

Round 1
Reviewer 1 Report
As the authors set forth in this paper, graphene has been adopted in the electronic devices for multiple applications. In addition, it can be with CMOS compatibility (Adv. Optical Mater. 2018, 6, 1800985). So what is the advantage of graphite flakes composite compared with graphene? I suggest the authors should state this point clearly.
Author Response
Rebuttal
Dear Editor,
Thank you very much for arranging the review and valuable comments therein. We have addressed the comments and appropriate changes in the manuscript have been made accordingly. The changes are detailed below.
Comments from the reviewers:
As the authors set forth in this paper, graphene has been adopted in the electronic devices for multiple applications. In addition, it can be with CMOS compatibility (Adv. Optical Mater. 2018, 6, 1800985). So what is the advantage of graphite flakes composite compared with graphene? I suggest the authors should state this point clearly.
Thank you for suggesting the article. We had added it to our manuscript in Introduction section second last paragraph as,
“In [35], 2D materials and colloidal quantum dots graphene-based infrared photodetectors as CMOS for light harvesting. It has a gate-tuneable ambipolar feature gate bias <3.3 V and can detect spectra from visible to near-infrared light with gain up to 105 and response times of 3 ms. The phototransistor has 104 AW−1 responsivity and 1012 Jones-specific detectivity (at 1550 nm at 1V).
As literature presents that carbon have different application, in this study we will be more focused towards graphite. In simplistic terms, graphene is a monolayer of graphite, a mineral ubiquitous in nature and comprises numerous layers of graphene. Henceforth, the present article has opted for the utilization of graphite flakes owing to the presence of numerous graphene layers, which provides an opportunity for enhanced responsivity of the devices across a diverse range of force, acceleration, and thermal conditions.” For readers better understanding .
We thank the reviewers for improving our concepts and allowing us to highlight the corrections in the paper. We hope the revised manuscript will be accepted for publication.
On behalf of all the authors
Yours Sincerely,
Engr Dr Noshin Fatima
Assistant Professor
Faculty of Engineering, Technology and Built Environment,
UCSI University, Kuala Lumpur, 56000, Malaysia

Reviewer 2 Report
The authors investigated the physical properties of a graphite-based composite that is cheap and environmentally friendly. Although graphite-based structures have been extensively studied in previous literatures, the authors still conducted some novel experiments to show more details of their properties that are less known before. On the other hand, as they mentioned, these devices may be used in university research in international poverty line nations benefitted from their low costs, which is a good desire. Therefore, we recommend to consider publishing this manuscript after addressing the following comments.
1. Please add lattice structure sketches for the composite in Figure 1.
2. Please consider removing some equations that are too elementary to mention, such as the Newton's second law in (1) and the linear function model in (9).
The English language used in the manuscript is grammarly correct, but not native. The authors are highly encouraged to find native speakers and improve the language style.
Author Response
Rebuttal
Dear Editor,
Thank you very much for arranging the review and valuable comments therein. We have addressed the comments, and appropriate changes in the manuscript have been made accordingly. The changes are detailed below.
Comments from the reviewers:
The authors investigated the physical properties of a graphite-based composite that is cheap and environmentally friendly. Although graphite-based structures have been extensively studied in previous literatures, the authors still conducted some novel experiments to show more details of their properties that are less known before. On the other hand, as they mentioned, these devices may be used in university research in international poverty line nations benefitted from their low costs, which is a good desire. Therefore, we recommend to consider publishing this manuscript after addressing the following comments.
Comments to the Author(s)
- Please add lattice structure sketches for the composite in Figure 1.
Thank you for the suggestion, but we are from an engineering background. We studied latice structure sketches about graphite which presents the layered structure of graphene. Regrettably, owing to a time constraint, the inclusion of the figure is presently unfeasible as we are unable to substantiate its validity. In due course, we shall take into consideration the inclusion of lattice structure illustrations to enhance the elucidation of our research.
- Please consider removing some equations that are too elementary to mention, such as the Newton's second law in (1) and the linear function model in (9).
Both equations are removed as per suggestion.
The paper “Fabrication and investigation of graphite flakes composite based non -invasive flex multi functional force, acceleration and thermal sensor” sounds interesting from the applications point of view From experimental point of view, is interesting and can be accepted for publication provide the author comply the above the following major revisions:
We thank the reviewers for improving our concepts and allowing us to highlight the corrections in the paper. We hope the revised manuscript will be accepted for publication.
On behalf of all the authors
Yours Sincerely,
Engr Dr Noshin Fatima
Assistant Professor
Faculty of Engineering, Technology and Built Environment,
UCSI University, Kuala Lumpur, 56000, Malaysia

Reviewer 3 Report
The paper “Fabrication and investigation of graphite flakes composite based non -invasive flex multi functional force, acceleration and thermal sensor” sounds interesting from the applications point of view
From experimental point of view, is interesting and can be accepted for publication provide the author comply the above the following major revisions:
- Please modify improving abstract with more details on: “thin-film multi-functional non-invasive sensor.”
- Explain better which kind of analyse
- Line 48:add citation
- Line 95 add “ranging from”
- 104 optical image?secondary electron?@15 kV?-----Scanning electron microscope
- Line 140 remove exp 3
- Please describe in the detaill the method of estimation of crystal lattice
- Line 149 insert deginstead of °
- Line 150 Cristallographic directio ,please insert bracket (011 )
- Line 153 X axes? /cm?
- Figure XRD,seems to noise in order in the 210 and 201 crystallographic direction assigment,please improve the signal to noise ratio,with new image
- Too many self citation,please replase it with others
Author Response
Rebuttal
Dear Editor,
Thank you very much for arranging the review and valuable comments therein. We have addressed the comments and appropriate changes in the manuscript have been made accordingly. The changes are detailed below.
Comments from the reviewers:
major revisions:
Comments to the Author(s)
- Please modify improving abstract with more details on: “thin-film multi-functional non-invasive sensor.” Explain better which kind of analysis.
Thank you for highlighting this point. The abstract is edited to highlight the morphological and electrical analysis in the abstract section.
- Line 48:add citation
Citation [14] added.
- Line 95 add “ranging from.
Done.
- 104 optical image?secondary electron?@15 kV?-----Scanning electron microscope
Thank you for the correction; we made the changes as per required. We initially added an optical image, which was later replaced with SEM.
- Line 140 remove exp 3
Sorry, we didn’t get it. Does it mean we have to remove expression 3.
- Please describe in the detaill the method of estimation of crystal lattice.
The crystal lattice was calculated by using Bragg’s law. Where the lambda was 0.15406 nm, (hkl) was (002), sin theta was 0.2242. crystal lattice =(0.15406*(SQRT((0^2)+(0^2)+(2^2))))/(2*0.2242)= 0.68 Å.
- Line 149 insert deginstead of °
As per the suggestion, the appropriate degree symbol has been incorporated in two places (Line 145 and Line 147, respectively).
- Line 150 Cristallographic directio ,please insert bracket (011 )
Corrected, thank you for the suggestion.
- Line 153 X axes? /cm?
Yes, the wavelength is calculated per centimeter. So we can write it in both ways i.e., /cm or cm-1.
- Figure XRD,seems to noise in order in the 210 and 201 crystallographic direction assigment, please improve the signal to noise ratio,with new image
The non-crystalline structure of isoprene sulfone posed a challenge in obtaining improved outcomes. The consistency of the peak values at 210 and 201 led us to conclude that they were not indicative of noise signals.
- Too many self citation,please replase it with others
The citations are reduced to 7 out of 47 (14%).
We thank the reviewers for improving our concepts and allowing us to highlight the corrections in the paper. We hope the revised manuscript will be accepted for publication.
On behalf of all the authors
Yours Sincerely,
Engr Dr Noshin Fatima (Assistant Professor)
Faculty of Engineering, Technology and Built Environment,
UCSI University, Kuala Lumpur, 56000, Malaysia

Round 2
Reviewer 1 Report
I suggest to accept this manuscript.
Reviewer 3 Report
Dear Author,
thanks for the revided appreciated work.
For the XRD if you performed several analyses confirming the same peaks,the paper for me is it ok for the pubblication.
Best Regards